# Inhibitory Effect of Phosphorothioate Oligonucleotide Complementary to G6PD mRNA on Murine Melanoma

**Kseniya A. Yurchenko** [1,*], **Kateryna V. Laikova** [2], **Ilya O. Golovkin** [2], **Ilya A. Novikov** [1], **Alyona A. Yurchenko** [1], **Tatyana P. Makalish** [2] and **Volodymyr V. Oberemok** [1]

1   Department of Molecular Genetics and Biotechnologies, Institute of Biochemical Technologies, Ecology and Pharmacy, V.I. Vernadsky Crimean Federal University, 295007 Simferopol, Russia
2   Medical Academy Named after S.I. Georgievsky, V.I. Vernadsky Crimean Federal University, 295007 Simferopol, Russia
*   Correspondence: cfuv@crimeaedu.ru

**Abstract:** In terms of the incidence among all tumors, skin cancer is on top, with the most deadly among them being melanoma. The search for new therapeutic agents to combat melanoma is very relevant. In our opinion, antisense oligonucleotides (ASO) aimed at suppressing the genes responsible for their viability in cancer cells give hope for treatment, which makes it possible to eliminate cancer cells near the tumor site both before and after surgery. In this article, we describe how Skeen-11 phosphorothioate oligonucleotide significantly decreased the proliferative activity of murine melanoma cells. Injections of Skeen-11 also inhibited tumor growth in mice with inoculated melanoma. A toxicity study showed no side effects with dose adjustments. The results show that the use of ASO Skeen-11 in vivo reduced the tumor size within 7 days, reduced the number of mitoses in the tumor cells, and increased the amount of necrosis compared with the control group.

**Keywords:** melanoma; glucose-6-phosphate dehydrogenase; phosphorothioate oligonucleotides; cell proliferation; cancer

## 1. Introduction

Cutaneous malignant melanoma is one of the most aggressive malignancies, characterized by a strong propensity to metastasize and the lowest survival rate [1]. According to the World Health Organization (WHO), for every three patients diagnosed with cancer, one will have skin cancer [2]. Although melanoma constitutes ~5% of all skin cancers, it accounts for >75% of skin cancer deaths [3].

To treat melanomas by removing local micrometastases, a wide excision is used along with normal tissue. This happens because normal tissue can hide genotypically abnormal cells in the surrounding skin or superficial lymphatics [4]. The balance between removing cancer cells and not causing functional and cosmetic harm to the patient is difficult to find. Time is of the essence when it comes to skin cancer. If detected early, many melanomas can only be treated with surgery. By themselves, melanoma cells are highly invasive, rapidly metastasize, and when the disease progresses, the survival prognosis is unfavorable [5].

Addressing therapeutic options, such as finding new targets for the treatment of metastatic melanoma, remains a challenge but is urgently needed [6]. Most of what is currently understood about cancer evolution has come from observing cellular changes in tumor biopsies or alterations in DNA from tumor cells [7]. Cancer cell lines can be readily used for drug discovery and connecting genomic alterations to drug responses [8]. However, certain in vitro experiments are often difficult or impossible to replicate in vivo and the suitability of cell lines as tumor models has been questioned [9]. Thus, there is a need to test the data obtained from cancer cell lines in an animal model.

The main characteristics of many cancers are processes associated with energy metabolism, such as glycolysis and increased glucose uptake, as well as a decrease in oxidative metabolism.

Along with glycolysis, the pentose phosphate pathway is also an important metabolic pathway. The pentose phosphate pathway generates pentose phosphates, which maintain a high rate of nucleic acid synthesis, which is critical for cancer cells. It also provides NADPH, essential for both fatty acid synthesis and cell survival under stressful conditions with high levels of intracellular reactive oxygen species. As with glycolysis, most enzymes of the pentose pathway in cancer are not regulated [10,11].

One of the central enzymes of the pentose phosphate pathway (PPP) is glucose-6-phosphate dehydrogenase (G6PD) [12]. Recently, research with sesquiterpenoid polydactin (a natural molecule found in *Polygonum cuspidatum* and other plants) demonstrated that polydactin directly inhibits G6PD, causing redox imbalance. This, in turn, results in endoplasmic reticulum stress, cell cycle arrest, and apoptosis and causes a significant decrease in cancer cell survival, along with a decrease in NADPH levels [13]. Data from an in vitro cellular study suggested that G6PD suppression impaired cell migration, invasion, and epithelial-mesenchymal transition. In addition, G6PD knockdown activated the JNK pathway, which then blocked the AKT/GSK-3β/Snail axis to induce E-Cadherin expression and transcriptionally regulated MGAT3 expression, thus promoting bisecting GlcNAc-branched N-glycosylation of E-Cadherin [14]. In addition, it was shown that the expression and activity of G6PD positively correlates with the expression of mRNA and proteins of cyclins D1 and E, p53, and S100A4. These data suggest that G6PD can regulate the cell cycle through its influence on these factors, thus indirectly regulating melanoma growth. The increased expression of p53 observed in vitro in A375 cells correlated well with high G6PD activity, suggesting that the p53 protein is stimulated by G6PD [13]. A decrease in the G6PD level in melanoma cell lines leads to a decrease in phosphorylated STAT5, which is important for uncontrolled cell growth. Protein expression at a STAT3/5 ratio and a phosphorylated STAT3/5 ratio is reduced in G6PD-deficient melanoma cells. The absence of G6PD in mouse melanoma cells enhances apoptosis by increasing the FAS level and decreasing the Bcl-2 and Bcl-xL levels [15]. Thus, G6PD plays an extremely important role in the management of melanoma cells, participating in many cascades of biochemical reactions including anti-apoptosis and fast growth, which enable their aggressive nature.

Currently, a promising direction for the treatment of various diseases is the use of drugs based on antisense technologies. Several drugs based on antisense technologies (mipomersen, nusinersen, eteplirsen, etc.) have already been developed and approved for use [16,17]; however, while several more are undergoing clinical trials [18], there are no such drugs on the market for cancer. The active study of the pathogenesis of various diseases, including cancer, and the molecular pathways underlying them, opens up many targets for therapeutic intervention. A great advantage of antisense technologies lies in their specificity, since antisense oligonucleotides (ASO) selectively block or destroy the target mRNA, thereby reducing protein expression [18,19]. Correct selection of the length and modifications of ASOs results in high efficiency and stability [20], and in our previous work with insects and rheumatoid arthritis, 11-mer ASOs were shown to be the most optimal and effective [21–23]. For this reason, we decided to synthesize the 11-mer ASO Skeen-11, aimed at suppressing the expression of G6PD, and study its effect on the growth and development of cutaneous melanomas in mice.

## 2. Results

Our experiments demonstrated the effectiveness of ASO Skeen-11, which led to a significant decrease in the cell index of cancer cells in a cell culture model murine with a melanoma cell line. The most pronounced effect was observed 6 h after the beginning of the experiment, where the cell index (proliferation) of the culture treated with Skeen-11 was $0.35 \pm 0.02$; within 6 h, this value had decreased dramatically to $0.18 \pm 0.02$. In the control group, the cell index was $0.52 \pm 0.03$ after 6 h; after 12 h, it reached a growth plateau with a value of $0.48 \pm 0.03$. The mean decrease in the cell index at 12 h in the group treated with Skeen-11 was 2.67 times lower compared with that of the control group.

The random oligonucleotide OligoA-11 did not show any significant effect on melanoma growth compared to the control group (Figure 1e).

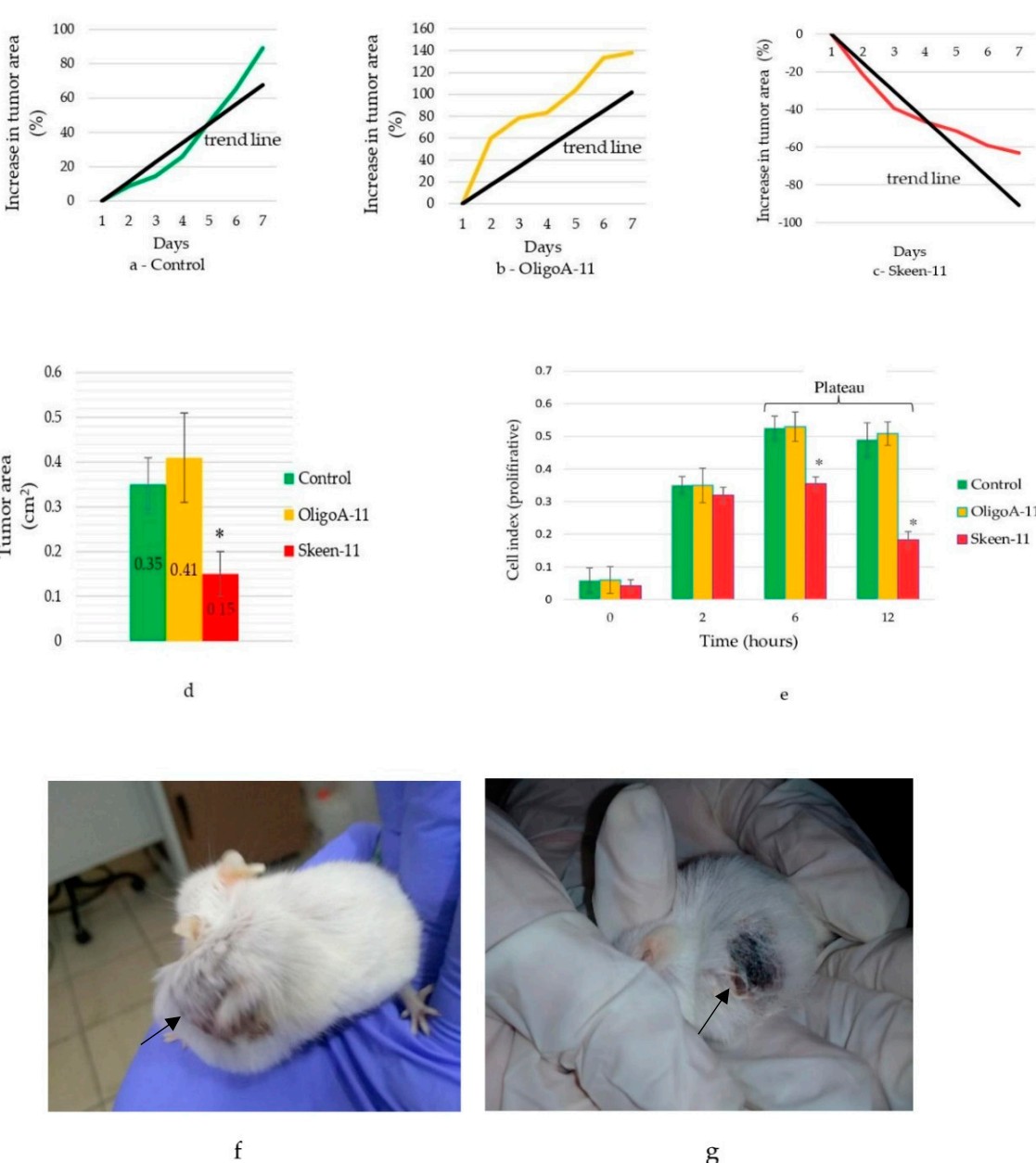

**Figure 1.** Dynamics of the average cumulative increase in tumor area for the different experimental groups on the 7th day of the experiment: (**a**)—control; (**b**)—OligoA-11; (**c**)—Skeen-11; (**d**)—average tumor area; (**e**)—dynamics of the cell index (proliferation) in different experimental groups; (**f**)—tumor formed on the 18th day after inoculation; (**g**)—tumor reduction on the 7th day after Skeen-11 treatment at a concentration of 30,000 ng/μL (21st day from the start of the experiment) (indicated by arrow); values are shown with the means and SE. A significant difference compared to the control group ($p < 0.05$) is marked with *.

To assess the effect of Skeen-11 in a mouse model, melanoma cells were injected sub-cutaneously (Figure 1f,g). During the experiment, a decrease in tumor size was found in response to the use of Skeen-11. The Skeen-11 experimental group treated for 7 days showed an average 15% reduction in tumor size (Figure 1c), while the control group had an increase in tumor size by 6% per day (Figure 1a). At the beginning of the experiment, the average size of the tumors in the groups was $0.29 \pm 0.04$ cm$^2$; on the 7th day of

the experiment, there were significant differences between the average tumor area of the control group ($0.35 \pm 0.06$ cm$^2$) and the group receiving Skeen-11 ($0.15 \pm 0.05$ cm$^2$) ($p < 0.05$) (Figure 1d). Interestingly, no tumor growth was seen at the Skeesn-11 injection site (the tumor continued to grow in the opposite direction from the injection site). The control oligonucleotide OligoA-11, compared with the untreated control, did not significantly affect the growth of melanoma ($0.4 \pm 0.11$ cm$^2$).

On the 7th day, the tumor area in the Skeen-11 group had decreased by 51%; in the control group, it had increased by 20.7% compared to the first day of the experiment.

Interestingly, when treatment with Skeen-11 was discontinued on the 8th day, the tumor area in the treated animals reached the size of those in the controls in 2 days, proving that Skeen-11 inhibited the melanoma cells from aggressive growth.

The results of an immunohistological examination for the Bcl-2 marker showed that for the control group, $7 \pm 0.47$ cells were found in the field of view ($p < 0.01$), while in the treatment group, $11.8 \pm 0.84$ cells were found, which is 40.67% more. According to the FAS marker, $3.6 \pm 0.45$ cells were found in the field of view in the control group ($p < 0.01$), and in the treatment group, $10.2 \pm 1.06$ cells were found, which is 64.7% more. Interestingly, 2.5 times less amitoses were observed in the Skeen-11 group compared to the control group ($p < 0.05$). In addition, the vacuolization of nuclei in tumor cells located near the vessels is noteworthy and there are signs of cell dystrophy (Figure 2a).

The apoptotic effect of ASO Skeen-11 was studied on Clone M-3 murine melanoma cell line, which are characterized by highly aggressive tumor growth. The apoptotic effect of Skeen-11 ASO was studied by staining CloneM-3 using the kit «Annexin V and Dead Cell» followed by flow cytometry.

It was shown that under the action of Skeen-11 as a specific ASO, after 12 h of incubation, an increase in the percentage of Clone M-3 cells with signs of apoptosis to $96.1 \pm 6.88\%$ ($p < 0.05$) was observed, which is significant compared to the control—$9.01 \pm 2.68\%$ (Figure 2($b_1$,$b_2$)) ($p > 0.05$).

In the Skeen-11 group, the migration activity of Clone M-3 cells was suppressed by the "overgrowth of a scratch" method (scratch test) (Figure 2d).

As can be seen from the presented data, the phosphorothioate oligonucleotide Skeen-11 significantly reduces the rate of melanoma cell migration. The rate of scratch healing in the control was $14.3\% \pm 3.3$ after 24 h and $17.3\% \pm 3.3$ after 72 h of incubation, while Skeen-11 contributed to the destruction of the cell monolayer from the edges of the scratch outward, after 24 h by $13.3 \pm 3.3$ and after 72 h by $23.3 \pm 3.3\%$, which is significant compared with the control ($p < 0.05$).

The toxicity of ASO was investigated by biochemical analysis of the blood of mice, hematological analysis, and immunohistological analysis.

The toxicity of phosphorothioate oligonucleotides was assessed by morphometric analysis of destructive changes in the liver and kidneys of animal tumor carriers. It has been shown that the development of a tumor in itself has a toxic effect on the liver of tumor-bearing animals, which is expressed in destructive changes in liver tissues, such as dystrophy and apoptosis, reaching 15% of the entire parenchyma of the organ. The introduction of phosphorothioate oligonucleotides does not have significant additional destructive effects on the liver and kidneys of animals.

The liver of the control group has a typical structure. The area of hepatocytes is 83.5%. Hepatocytes are organized into beams and form hepatic lobules with central veins in the center and triads along the periphery. Hepatocytes have a rounded or multifaceted shape with a fine-grained cytoplasm and a rounded nucleus with well-defined nucleoli. In the OligoA-11 group, hypertrophy of hepatocytes (85% area) and their nuclei were observed, and polyploid nuclei were often found. The organization of hepatocytes within the liver lobules is impaired. In the Skeen-11 group, the sinus capillaries are dilated, which somewhat complicates blood filtration. Focal lymphoid infiltration of the parenchyma was observed, however, to a significantly lesser extent than in the OligoA-11 group. There were also individual hypertrophied hepatocytes, the area of hepatocytes was 80% with

polyploid nuclei, and a large number of binuclear cells were present, which indicates active regeneration processes. The structure of the lobules was not disturbed.

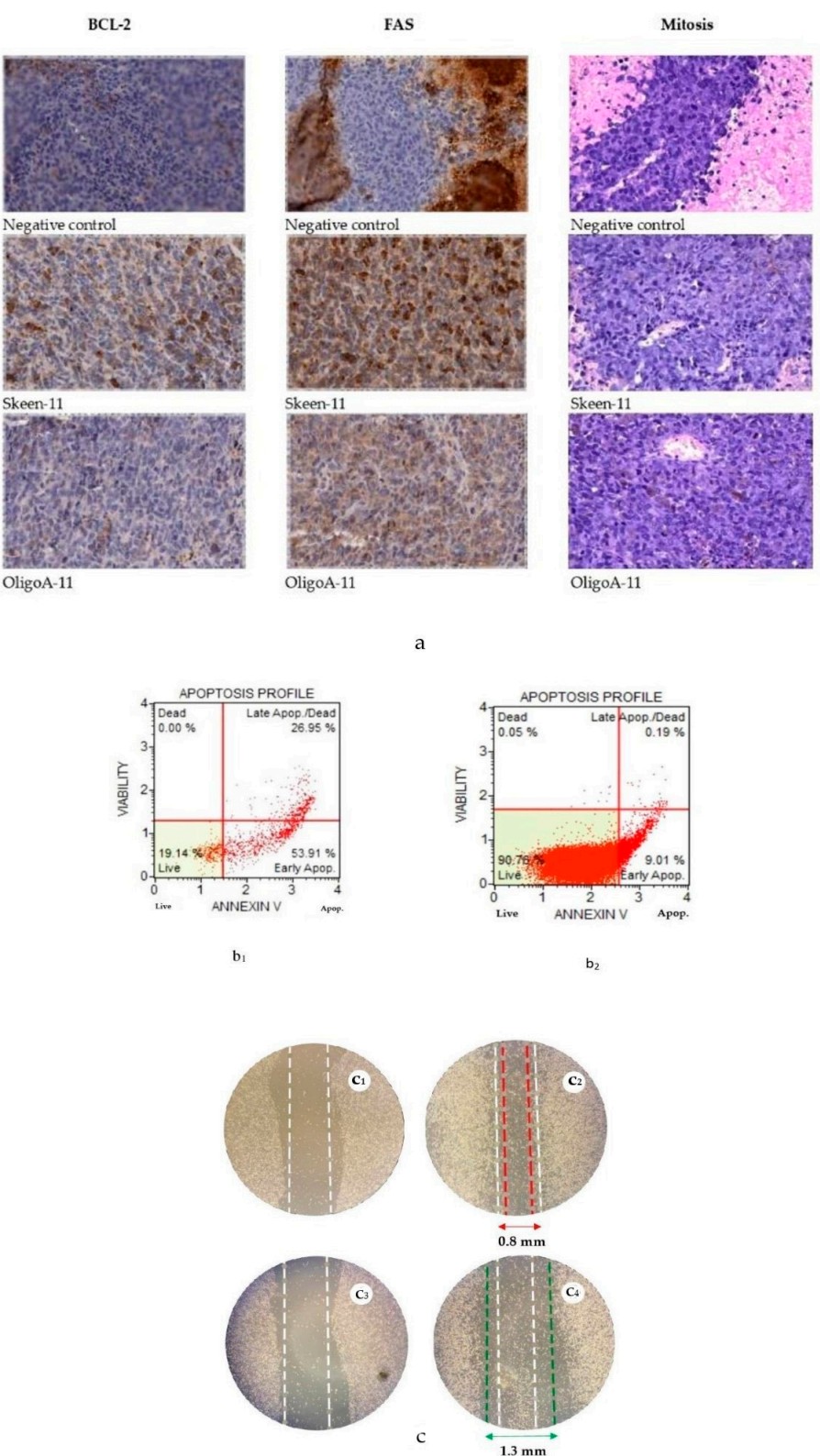

**Figure 2.** (**a**) Histological sections of melanoma tumors in different groups of the experiment (control, Skeen-11, OligoA-11) (magnification 400×). (**b**) Apoptosis profile of melanoma cells from different groups, stained with Annexin V and Dead Cell 12 h after exposure to: (**b₁**)—Skeen–11; (**b₂**)—control.

(**c**)—scratch-test: control—(**c$_1$**) 1th day of the experiment, scratch = 1 mm; (**c$_2$**) 3th day of the experiment, scratch = 0.8 mm; Skeen-11—(**c$_3$**) 1th day of the experiment, scratch = 1 mm; (**c$_4$**) 3th day of the experiment, scratch = 1.3 mm; cell photography was performed using a microscope every 24 h for 3 days; values are shown with the means and SE.

Thus, in the Skeen-11 group, the liver has some signs of damage and repair, while the differences from the control group are insignificant and concern only the area of nuclei and the nuclear–cytoplasmic ratio. In the OligoA-11 group, hepatocytes are severely damaged, and reparative processes cannot compensate for these damages. As a consequence, this leads to cell death via the apoptotic pathway and elimination of apoptotic cells by leukocytes, as evidenced by extensive areas of leukocyte infiltration. The kidneys of animals in the control group have a typical structure. In the experimental groups, in relation to the control group, there is an increase in the area of the renal corpuscle and an increase in the external dimensions of the proximal tubules due to the height of the epithelium, which significantly differs from the control in the Skeen-11 group, which may indicate compensatory changes due to active filtration of primary urine and impaired reabsorption of organic substances. In the same group, the lumen of the distal tubule and collecting ducts was enlarged. In the OligoA-11 group, the sizes of individual elements of the nephron were also increased, but to a much lesser extent. At the same time, cell damage is much more pronounced, on the basis of which we can assume a higher toxicity of the substance (Figure 3a–g).

The assessment of blood biochemistry was carried out according to five main biochemical parameters of blood, which show the state and functional work of internal organs, in particular, the functionality of the liver and kidneys. Comparative analysis does not show significant differences between the experimental and control group. Untreated mice with tumors were taken as a negative control. Model animals that are healthy and without tumors are considered positive controls. The results of the positive control assay for mice were considered the reference.

An analysis of the content of erythrocytes (RBC) in the blood of different experimental groups showed that these blood elements are within the normal range. No significant differences were found (Figure 3h). In the study of thrombocytes (PLT) in different experimental groups, it was found that the thrombocytes count in the Skeen-11 and OligoA-11 groups was increased by 237 units compared to the positive control and 249 units accordingly, this is due to the fact that with targeted treatment at high doses, an increase in thrombocytes is observed [24]. In the negative control the indicator is increased by 201 units, since it is known during the cancer number of thrombocytes is increased [25] (Figure 3i). WBC testing showed an increase in the negative control compared to the positive control, Skeen-11, and OligoA-11 groups, which is explained by the presence of a tumor and an inflammatory process. Accordingly, in OligoA-11, an increase in leukocytes is also associated with the presence of a tumor and its progression.

Significant differences are observed in the negative control and OligoA-11 (Figure 3j).

In general, Skeen-11 corrected blood biochemical parameters towards the positive control, except the ALAT parameter. At the same time, the indicator differed most from the norm (positive control) according to ALAT. However, it is known from the literature that ASOs can have moderate hepatotoxicity due to binding to certain proteins [26]. All this can be corrected with the right dose [27]. Thus, Skeen-11 improved blood chemistry compared to the control OligoA-11 oligonucleotide and compared to the negative control (Figure 3k).

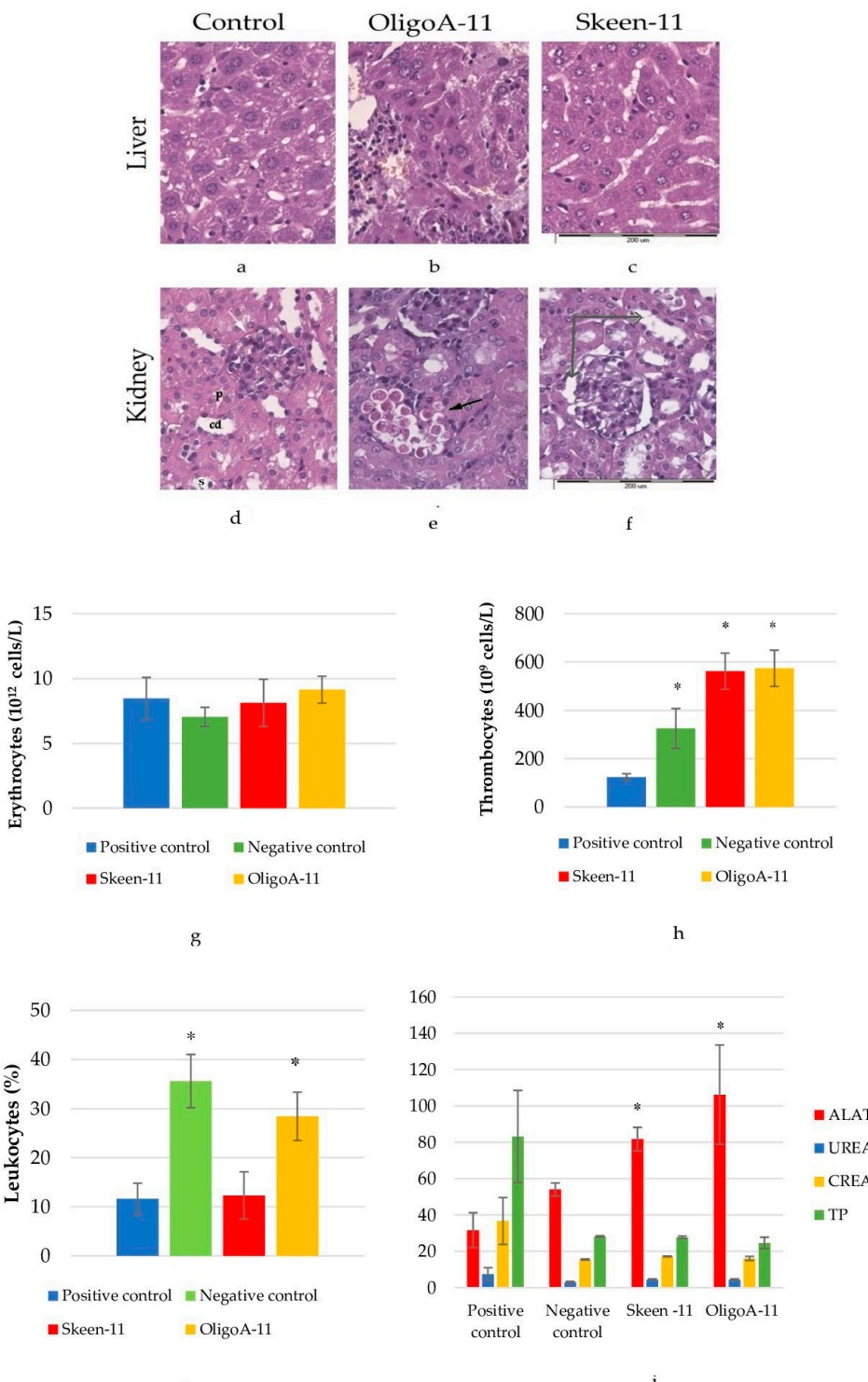

**Figure 3.** Mouse liver. Hematoxylin-eosin. Approximately 40×. (**a**)—Control group, (**b**)—OligoA-11 group, (**c**)—Skeen-11 group; cortex of the kidney. Hematoxylin-eosin. Approximately 40×; (**d**)—Control group(p—proximal tubule, s—distal tubule, cd—collecting duct); (**e**)—OligoA-11 group (black arrow—desquamation of proximal tubule epithelium and accumulation of lymphocytes next to it); (**f**)—Skeen-11 group (gray arrow is the dilated urinary space of the renal corpuscle and the lumen of the distal tubule); (**g**)—analysis of erythrocytes; (**h**)—analysis of thrombocytes; (**i**)—analysis of leukocytes; (**j**)—biochemical parameters of mice blood after ASO therapy on the 8th day from the start of the experiment (statistics performed compared with negative control); values are shown with the means and SE. A significant difference compared to the control group ($p < 0.05$) is marked with *.

## 3. Discussion

The results of these studies show the promise of using phosphorothioate oligonucleotides for blocking melanoma cell division. A single use of oligonucleotides for 12 h in cell culture reduced the proliferative activity by 62%, and daily injection of oligonucleotides into inoculated melanoma mice led to a decrease in tumor size of 57% compared to tumor size in the control group. Immunohistological staining for markers Bcl-2 and FAS produced an interesting result. It is known that during malignant transformation, the expression of these markers decreases. Subsequently, a decrease in Bcl-2 expression makes the tumor less attractive to the immune system [28,29]. The FAS receptor plays an important role in triggering the apoptosis mechanism externally through the activation of caspases [30,31]. Thus, a decrease in the expression of FAS and Bcl-2 worsens the treatment prognosis. In our experiment, during immunohistological staining in the treatment group, a more intense staining for markers FAS and BCL-2 was noted, which in turn indicates an increased expression of these markers and a more favorable treatment prognosis. Separately, it should be noted that during the histological study, a significant decrease in the number of amitoses in the treatment group compared with the control group was noted, along with observation of necrosis of newly formed cells.

The discovered apoptotic effect on PPP from the use of Skeen-11 was also recorded earlier. It has recently been shown that it blocks G6PD causing accumulation of reactive oxygen species and a strong increase in endoplasmic reticulum stress. After these processes, there is a blockage of the cell cycle in the S-phase, about 50% apoptosis, and 60% inhibition of the invasion of cancer of the tongue in vitro [12]. In our experiment, the apoptosis pathway for melanoma cells resulted in tumor reduction in model animals.

The results presented here do not directly explain the mechanisms of hepatotoxicity. However, we suggest that the hepatotoxic potential is related to the propensity of the oligonucleotide to bind to certain proteins. It is well known that oligonucleotides can interact with specific proteins [27], and these interactions can be sequence dependent, as is the case with Toll-like receptors binding oligonucleotides containing cg dinucleotides [32].

The most frequently reported serious dose-limiting acute toxicities of phosphorothioate ASOs administration are the transient activation of the complement cascade, prolongation of the partial thromboplastin time (PTT), thrombocytopenia, and elevated serum transaminases. Most of these toxicities result from non-specific interactions between ASO and plasma proteins [33,34].

Limiting toxicity is associated with hepatocellular degeneration leading to a decrease in total protein levels. Despite this toxicity, which is generally mild and easily treated with available standard drugs, clinical trials have clearly shown that ASO can be administered safely to patients [35]. It is important to note that phosphorothioate ASOs are generally safe at therapeutic doses. ASO-induced toxicity occurs at a dose higher than the therapeutic dose commonly used in clinical trials [28].

In summary, current evidence shows limited untoward effects and reversibility of the damage induced phosphorothioate ASOs, with promising effectiveness for the treatment of various pathologies [36].

Thus, inhibition of G6PD expression with antisense oligonucleotides is an effective tool for the treatment of melanoma, but it should be noted that this method is most suitable for local treatment, since central administration leads to a large number of side effects. It is predictable that G6PD, an important enzyme for actively dividing cells, will, when used centrally, affect the hematopoietic, reproductive, and other systems, which does not make it better than chemotherapy. However, such ASOs can be very useful in combination therapy. For example, after removal of the tumor, to reduce the risk of recurrence, the edges of the wound could be treated to 'finish off' the escaped tumor cells. We plan to include the ASO in the composition of an ointment to see how effective this delivery method is. If we find that tissue penetration is poor because ASOs are negatively charged, they can easily be delivered using electrophoresis.

Taking into account the study of cell proliferative activity, histological, apoptotic, and toxicity studies, Skeen-11 is a promising anti-melanoma phosphorothioate ASO sequence that can be used to reduce melanoma cell proliferation After several surgeries, melanoma deaths are the most common [37–39]. In our opinion, antisense oligonucleotides, which can be used in the form of targeted ointments or creams [5], or injections, as shown in this article, provide particular hope for the elimination of cancer cells in the area near the tumor focus, both before and after surgery, to delay or even prevent the primary tumor from developing metastases and sending them to the lymph nodes.

## 4. Materials and Methods

Tumor cells

The study used cancer cells from the Clone M-3 murine melanoma cell. Cells were cultured in DMEM F-12 nutrient medium supplemented with 1% penicillin/streptomycin antibiotics, 1% pyruvic acid, and 10% fetal bovine serum (FBS). Cell transplantation was performed using Dulbecco's phosphate-buffered saline (DPBS) and trypsinization, followed by neutralization of the trypsin with the prepared nutrient medium.

The determination of the cell index was carried out in 12 replicates from experiments performed in real time for 24 h in xCELLigence RTCA DP Analyser (ACEA Biosciences, USA). For the experiment, 6000 cells were added to each well of the RTCA DP Analyzer.

Animals and treatment groups

A simple, blind, prospective, randomized, comparative study was conducted in BALB/c mice. The animals were divided into 3 groups, each of which consisted of 40 individuals. Each group was transplanted with melanoma cells. The first group—control—was treated with 0.9% NaCl; the second group—OligoA-11 (5′-AAA-AAA-AAA-AA-3′) —was treated with a control oligonucleotide at a concentration of 30,000 ng/μL; and the third group—Skeen-11—was treated with an experimental antisense oligonucleotide (5′-CTG-AAT-CTC-CG-3′; GenBank: NM_008062.3) at a concentration of 30,000 ng/μL. All injections of phosphorothioate oligonucleotides were performed at a distance of 1–1.5 cm from the site from the edge of the tumor. The injection volume was 100 μL.

Design and synthesis of antisense oligonucleotides

The design of phosphorothioate antisense oligonucleotides was done independently based on the mouse genomic sequence available in the GenBank database (https://www.ncbi.nlm.nih.gov/genbank). Accessed on 18 February 2020.

Synthesis of phosphorothioate antisense oligonucleotides was carried out using the standard phosphoramidite method. The correspondence of the synthesized DNA fragments was determined using a BactoSCREEN analyzer based on a MALDI-TOF mass spectrometer (Litech, Moscow, Russia).

Mice and transplantation of cancer cells

One hundred and twenty mice (12–18 g) from an inbred, non-immunosuppressive line of BALB/c mice were selected for the experiment.

Preparation of cells for transplantation into the mice was carried out as follows: cells were removed and counted, and then centrifuged. The resultant pellet ($2 \times 10^5$ cells) was resuspended in 500 μL Hank's solution. Aliquots of the prepared suspension of melanoma cells were injected subcutaneously into the interscapular region, with the syringe needle in a horizontal position.

On the 14–16th day after the introduction of the cancer cell suspension, the tumor tissue began to grow. The tumors were round, fleshy, and dark blue in color.

Apoptosis

Apoptotic processes in Clone M-3 cell culture were assessed using Muse™ Annexin V and Dead Cell reagents on a Guava Muse flow cytometer (Luminex, Austin, TX, USA). The experiment was carried out in 6 replicates. The total apoptosis in cells was assessed, after 4 h of exposure to antisense oligonucleotides Skeen-11 and OligoA-11 at a concentration of 20,000 ng/μL; the control group of cells was exposed to water. One hundred microliters of Annexin V and Dead Cell solution was added to an Eppendorf tube, and 100 μL of a

$5 \times 10^5$ cell suspension was added to it. After that, the solution is mixed and incubated at room temperature for 20 min. After this time, the resulting solution is sent to the analyzer to obtain results.

Histology and immunohistochemistry

On the 7th day, the animals were taken out of the experiment by decapitation under anesthesia. Part of the tumor was separated for histological examination. The material was fixed in 10% buffered formalin for 12 h, after which it was dehydrated and impregnated with formalin according to the standard method in a carousel-type histoprocessor. After embedding in blocks, sections were made 4 μm thick and stained with hematoxylin and eosin. Finished preparations were examined under a DM2000 microscope (Leica, Wetzlar, Germany). The average number of mitoses was counted in at least ten fields of view. To study the readiness of melanoma cells for apoptosis, the slides were stained in a Bond-MAX automatic immunohistotainer (Leica, Sydney, Australia) with antibodies to BCL-2 (dilution 1:200, Novocastra, UK) and FAS (dilution 1:200, abcam, Washington, DC, USA) according to the protocol recommended by the antibody manufacturer. The label was detected using the Bond Polymer Detection System. The number of positively stained cells was counted in at least 10 fields of view by calculating the arithmetic mean. Cells with cytoplasm stained with diaminenzidine were considered positively stained. At the same time, a comparison was made with preparations stained with hematoxylin and eosin to exclude from the count cells that have melanin pigment in the cytoplasm.

Scratch-test

To study the migration activity, Clone M-3 melanoma cells were planted in culture flasks in DMEM F-12 medium. The control group was treated with sterile water, ASO Skeen-11 was added to the experimental group. During the experiment, cells were incubated in 5% $CO_2$ at 37 °C. Two hours after Skeen-11 cells were treated, the cell monolayer was scratched. After 0, 24, and 72 h, the cells were photographed using an Olympus CKX53 microscope at $4\times$ magnification. Cell photographs were processed using the ImageJ program. The degree of scratch overgrowth was estimated by the formula $\upsilon = (1 - x) \times 100\%$, where x is the ratio of the scratch width after 24 and 72 h to the scratch width at 0 h.

Biochemical and hematological analysis

Hematological analysis was carried out according to the protocol of the Mindray BC-2800 Vet (China).

The mice were bled by decapitation into clean centrifuge tubes. The blood was placed in a thermostat at a temperature of 37 °C for 30–60 min, then transferred to the cold in the refrigerator for 1–2 h. Serum in the form of a clear liquid is separated from the blood clot. Serum isolation; the resulting fibrin clot was separated from the walls of the test tubes by circling with a glass rod or wire and was centrifuged at 1500–2500 rpm within 15–20 min.

Serum samples were prepared for further biochemical analysis for the following indicators: ALAT (Alanine aminotransferase), ASAT (Aspartate aminotransferase), UREA (Urea), CREA (Creatinine), TP (Total protein).

Statistical Analysis

All data obtained as a result of the study were subjected to statistical processing using the STATISTICA 10.0 program. The normality of the trait distribution was assessed using the Kolmogorov–Smirnov method. Student's *t*-test was used to calculate the comparison of means. Using descriptive statistics methods, we calculated the mean value of the trait and standard deviation. Differences were considered significant if the error probability was $p < 0.05$.

## 5. Conclusions

Melanoma remains a challenge for researchers; effective treatments for melanoma have not been sufficiently developed, and the prognosis of patients with melanoma remains unfavorable. As a result of the studies, it was shown that in vitro, ASO Skeen-11 significantly reduced the cell index in melanoma cells compared to that in the control group. In vivo, the use of ASO Skeen-11 reduced the tumor size within 7 days; when the treatment

was interrupted, the tumor size began to increase rapidly and reached the same size as that in the control group within 2 days. Histological staining demonstrated that the intensity of staining for markers FAS and Bcl-2 was higher in the treatment group than in the control group. It was also noted that the use of ASO Skeen-11 reduced the number of mitoses in the tumor cells and increased the amount of necrosis compared with the control group. Using Skeen-11 as an example, it was shown that phosphorotioate antisense oligonucleotides remain promising tools for blocking the proliferative activity of cancer cells and, at the right doses and with the right target gene, have a comprehensive and relatively safe effect on melanoma in vivo.

**Author Contributions:** Conceptualization, V.V.O.; methodology, K.A.Y. and V.V.O.; software, I.A.N.; formal analysis, I.O.G. and K.A.Y.; investigation, K.A.Y., A.A.Y. and T.P.M.; resources, I.A.N.; data curation, V.V.O. and K.V.L.; writing—original draft preparation, V.V.O. and K.A.Y.; writing—review and editing, V.V.O. and K.A.Y.; visualization, K.A.Y. and I.O.G.; supervision, V.V.O.; project administration, V.V.O. and K.V.L. All authors have read and agreed to the published version of the manuscript.

**Funding:** This research was funded by state assignment V.I. Vernadsky Crimean Federal University for 2021 and the planning period of 2022–2023 No. FZEG-2021-0009 ("Development of oligonucleotide constructs for making selective and highly effective drugs for medicine and agriculture", registration number 121102900145-0).

**Institutional Review Board Statement:** All manipulations with animals were carried out in accordance with the European Convention for the Protection of Vertebrate Animals Used for Experiments and Other Scientific Purposes and approved by the ethical committee of the Crimean Federal University (Protocol № 8 from 17 September 2020).

**Informed Consent Statement:** Not applicable.

**Data Availability Statement:** All relevant data that are not present is available from the authors.

**Acknowledgments:** We are grateful to the laboratory of DNA technologies of the Crimean Federal University for their contribution to the research.

**Conflicts of Interest:** The authors declare no conflict of interest. The funders had no role in the design of the study; in the collection, analyses, or interpretation of data; in the writing of the manuscript; or in the decision to publish the results.

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
