# Peer review of "Inhibitory Effect of Phosphorothioate Oligonucleotide Complementary to G6PD mRNA on Murine Melanoma"

_cimb, doi:10.3390/cimb45040207_

Round 1
Reviewer 1 Report
Thank you for good work
I recommended new title to be
Inhibitory effect of phosphorothioate oligonucleotide com3 plementary to G6PD mRNA on murine 4 melanoma
Refrecnces nedded to be updated till 2023
Thank you
Yours
Author Response
New title changed to “Inhibitory effect of phosphorothioate oligonucleotide complementary to G6PD mRNA on murine melanoma”
Added references 2023
Reviewer 2 Report
In the manuscript “Old tool for old target: phosphorothioate oligonucleotide complementary to G6PD mRNA shows inhibitory effect on murine melanoma”, Yurchenko et al described how antisense oligonucleotides (ASO) can significantly decrease the proliferative activity of murine melanoma cells. To suppress the gene responsible for their viability in cancer cells, glucose-6-phosphate dehydrogenase (G6PD) was selected as a target. G6PD plays an extremely important role in the management of melanoma cells, participating in many cascades of biochemical reactions including anti-apoptotosis and fast growth. Correct selection of the length and modifications of ASOs would result in high efficiency and stability. To selectively block or destroy this target mRNA, thereby reducing protein expression of G6PD, the authors used Skeen-11 phosphorothioate oligonucleotide, a 11-mer ASO, to study the effect of Skeen-11 on the growth and development of cutaneous melanomas in mice. As expected, injections of Skeen-11 inhibited tumor growth in mice with inoculated melanoma. In addition, a toxicity study seemed to show no significant side effects. Taking into account the collective results of cell proliferative activity, histological, and apoptotic studies, Skeen-11 is indeed a promising anti-melanoma phosphorothioate ASOs sequence for reducing melanoma cell proliferation. This report is very easy to read, well written, interesting, and useful for further medical applications. Accordingly, this reviewer recommends publication. Minor revision is needed for standardizing the type of references and remodeling the manuscript by using the manuscript template of this journal. In addition, the effect of other ASOs sequences targeting similar pathway or G6PD in melanoma cells for treatments should be discussed in this manuscript.
Author Response
No data were found on inhibition of G6PD ASOs, only another biological molecules
Reviewer 3 Report
I would like to suggest the following to make the following corrections: - to add more detailed information about “Skeen-11” (the structure/references); - in Introduction part the authors mention (line 88) “decided to synthesize the 11-mer ASO Skeen-11” but further no any synthetic part is described, this point should be specified; - the Abstract should be corrected by adding the specific results (for ex, from the conclusion part).
Author Response
All data about Skeen-11 added to materials and methods. Abstract was corrected.
Reviewer 4 Report
The authors report that they have experimented with treatment of melanoma of murine with antisense of G6PD with good results. The experiments are carefully performed and the results or their presentation may be adequate.
I would like to urge you to reconsider the following.
* The title "old tool for old target" is self-deprecating and literary, so how is it a good title for an academic paper?
* Regarding some of the points presented in Figure 3: the increased platelet count is also the impression that it occurs in ASO (including controls) (it is also elevated in the green negative control, which is this, should we take issue with it being low in the blue positive control?) Isn't this consideration necessary? 
* Similar to the above, the differences in WBC should be well considered.
* K in Figure 3, but the increase in ALAT is problematic, and shouldn't this change over time be presented?
Author Response
- New title changed to “Inhibitory effect of phosphorothioate oligonucleotide complementary to G6PD mRNA on murine melanoma”
- In the study of thrombocytes (PLT) in different experimental groups, it was found that the thrombocytes count in the Skeen-11 and OligoA-11 groups was increased by 237 units compared to the positive control and 249 units accordingly, this is due to the fact that with targeted treatment at high doses, an increase in thrombocytes is observed. In the negative control the indicator is increased by 201 units, since it is known that in cancerous tumors increase thrombocytes.
- WBC testing showed an increase in the negative control compared to the positive control, Skeen-11 and OligoA-11 groups, which is explained by the presence of a tumor and an inflammatory process. Accordingly, in A-11, an increase in leukocytes is also associated with the presence of a tumor and its progression.
- Taking into account the increased activity of ALAT in groups exposed to targeted therapy, this can be mistaken for drug-induced damage to liver cells. which may not have visible functional disorders from other organs. Based on the data on histochemical analysis, this is a reversible process, since the cells are at the stage of repair. Unfortunately, it was not possible to check the indicators over time, but only immediately after the experiment.